# Characterization of Microbial Diversity of Two Tomato Cultivars through Targeted Next-Generation Sequencing 16S rRNA and ITS Techniques

**DOI:** 10.3390/microorganisms11092337

**Published:** 2023-09-18

**Authors:** Rukayat Abiola Abdulsalam, Oluwatosin Ademola Ijabadeniyi, Errol D. Cason, Saheed Sabiu

**Affiliations:** 1Department of Biotechnology and Food Science, Durban University of Technology, Durban 4000, South Africa; 2Department of Animal Science, University of the Free State, Bloemfontein 9300, South Africa

**Keywords:** alpha- and beta-diversity, amplicon sequencing variants, high-throughput sequencing, jam tomato, microbial diversity, round tomato, spoilage organisms

## Abstract

Even though the nutritional and economic values of *Solanum lycopersicum* (tomato) are substantially impacted by microbial spoilage, the available data on its microbial community, particularly during spoilage, are limited and have primarily been characterized using conventional culture-dependent methods. This study employed a targeted high-throughput next-generation sequencing method to longitudinally characterize the microbial diversity of two South African tomato cultivars (jam and round) at varied storage intervals (1, 6, and 12 days). Throughout the storage period, the bacterial communities of the two cultivars were more diverse than the fungal communities. The microbial diversity of both bacteria and fungi was greater and comparable between the cultivars on day 1, but becomes distinct as the storage period increases, with round tomatoes being more diverse than jam tomato, though, on day 12, jam tomato develops greater diversity than round tomato. Overall, the most abundant phyla (though Proteobacteria was most dominant) were Proteobacteria, Firmicutes, and Bacteriodota in the bacterial communities, while Ascomycota and Basidiomycota formed most fungal communities with Ascomycota being dominant. At the genus level, *Pantoea* and *Klebsiella* (bacteria), *Hanseniaspora*, *Stemphylium*, and *Alternaria* (fungi) were prevalent. Taken together, this study casts light on a broad microbial diversity profile thus, confirms the cultivars’ diversity and abundance differences.

## 1. Introduction

*Solanum lycopersicum* L. commonly known as ‘tomato’, forms an integral part of the day-to-day diet worldwide probably due to its nutritional and health benefits [1]. Tomato is a flowery shrub plant that belongs to the family *Solanaceae*. It grows from 1–3 m (3–10 ft) in height and possesses a weak stem that often sprawls over the ground and often times may vine over other plants [2]. It is a seasonal crops with a limited lifespan of 140 to 168 days depending on the cultivars [3]. Botanically, tomato is referred to as a fruit [4] because its ovary contains seeds while nutritionally, they are classified as a vegetable; savoury with low amount of fructose and are served as part of the main course [4]. In 2020, Food and Agriculture Organization Corporate Statistical Database (FAOSTAT) reported a global tomato production of 186.821 million metric tonnes cultivated on 5,051,983 hectares of land yielding an average of 37.100 metric tonnes/hectare (mT/ha) [5]. As of 2020, China was the world’s largest tomato producer with 64.768 million mT, followed by India (20.573 million mT) and Turkey, ranked third with 13.204 million mT [4]. In Africa, Egypt produces 7.297 million mT of tomatoes in 2017, making it the continent’s leading producer; however, South Africa ranked seventh [6]. It is noteworthy that, South Africa’s annual tomato production is around 600,000 tonnes, accounting for about 24% of the total vegetable production in the country [3]. Jam (also known as roma), round (also known as beefsteak) and cherry tomato are the three commonly grown tomato in South Africa; however, the first two are mostly consumed [3]. Tomato plants are simple to grow and extremely productive, especially in the warm areas and/or seasons [4]. For example, tomatoes are grown in all South African provinces; however, Limpopo, Mpumalanga and Northwest provinces are best suited for tomato production due to their warm climate [4].

Nutritionally, tomato fruits are high in carotene, ascorbic acid, thiamine, riboflavin, and minerals [7]. It also contains bioactive substance such as quercetin, kaempferol, alpha-lipoic acid, cholin, naringenin, and lutein, as well as caffeic, ferulic, and chlorogenic acids [8]. The bioactive substance in tomato may have given it the potential to prevent degenerative diseases in humans, such as cardiovascular disease, cancer, and neurodegenerative disorders [9]. These constituents are important vitamins and minerals with established antioxidant attribute likely to afford it the potential to scavenge both free radicals and reactive oxygen species (implicated in several debilitating diseases), or by reducing cellular growth and damage, suppressing apoptosis, metal chelation, and signal transduction pathways [7,9]. Additionally, tomato production serves as a means of employment and lucrative source of income to farmers venturing into its cultivation. These same reasons could have been why tomato is among one of the most important vegetables with an annual value exceeding 90 billion USD [10,11]. Despite these benefits, tomatoes have a short postharvest shelf life, owing to the action of ethylene in the rapid ripening of the fruits, transpiration, senescence, and microbial spoilage, with the latter being the primary source of spoilage in tomato fruits [12]. Microbial spoilage of tomato is primarily caused by fungi and bacteria, with the former being the most implicated organisms [13,14]. In fact, annually, over 50% of tomato production has been reported to be lost due to microbial spoilage [14,15]. A good example of this was seen in December 2021, where the South African tomato industry recorded a loss of 94 million Rands majorly due to a destructive pest called *Tuta absoluta* [16].

Prior to the emergence of high-throughput sequencing (HTS) methods such as the Next Generation Sequencing (NGS) techniques, investigations into the microbial community structure of tomato fruits predominantly relied on culture-based methodologies [13]. The culture-based approaches involve isolation and subsequent identification of microorganisms using conventional molecular methods and this has identified spoilage fungal species such as *Aspergillus phoenicis*, *Absidia* sp., *Trichoderma* sp., *Alternaria alternata*, *Fusarium oxysporum*, *Aspergillus niger*, *Aspergillus flavus*, *Mucor* spp., *Rhizopus stolonifer*, *Botrytis cinerea*, *Penicillium* spp., *Geotrichum* sp., and *Phytophthora* sp. [17,18,19]. Additionally, bacterial isolates such as *Bacillus subtilis*, *Bacillus anthracis*, *Staphylococcus aureus*, *Bacillus cereus*, *Listeria monocytogenes*, and *Pseudomonas aeruginosa* have frequently been identified in deteriorated tomatoes through the conventional molecular methods [18,20]. However, culture-based techniques are time consuming and labour intensive, and quantifying the contribution of each organism to the microbial population structure in polymicrobial complex ecosystems is difficult [17]. Additionally, many organisms grow slowly while others cannot be cultured due to the unknown or unavailable essential growth requirements [18]. The introduction of culture-independent profiling methods for detecting fastidious or non-cultivable organisms by analysing the sequence of marker genes such as the 16S ribosomal RNA (16S rRNA) and Internal Transcribed Spacer (ITS) gene has sparked a revolution in biology and medicine, leading to the formation of the microbiome consortium [19,20,21]. Culture-independent characterization of bacterial and fungal communities can be achieved through amplification and sequencing of the 16S rRNA and ITS genes [22] or metagenomics approaches in which the sequences of the microbial community genes and genomes are obtained [23]. In general, 16S rRNA gene profiling is widely used because it is cost-effective and quick, and the results are tractable from an analytical perspective [24]. While studies have explored HTS methodologies to investigate the microbiota and mycobiota associated with tomato leaves [25], the rhizosphere of tomato plants [26], and compared the bacterial community structure of tomato fruit under varying cultivation conditions [27], no studies exist on the longitudinal characterization of tomato fruits including the ‘jam’ and ‘round’ cultivars, using NGS to date. It is on this background that the present study was designed to longitudinally (over the duration of spoilage) characterize both the microbiota and mycobiota from the ‘jam’ and ‘round’ tomato cultivars using 16S rRNA and ITS gene sequencing approaches. This is hoped to provide comprehensive and comparative baseline information on the microbial community of the two investigated cultivars which would be vital to future studies looking to establish the microbial structure of different species of tomato fruits.

## 2. Materials and Methods

### 2.1. Study Design

A longitudinal cohort study was conducted to characterize the microbial composition of two tomato cultivars (jam and round) commonly consumed in South Africa at different storage intervals. The entire processes were designed to simulate the natural tomato postharvest cultivation from farm to shelf until fork. 

### 2.2. Study Procedures

Two commonly eaten tomato varieties, ‘roma’ and ‘beefsteak’, commonly known as jam and round tomato, respectively, in South Africa, that are free of bruises were sourced at the time of harvest from a commercial farm in Durban, KwaZulu-Natal, South Africa on the 15 March 2022. Fifteen tomatoes (representative samples from top, middle, and bottom of the tomato crate) with clear, smooth, and firm texture were taken from individual varieties. To avoid contamination during handling and transportation, the tomatoes were kept in a polyethylene terephthalate (PET) container directly from the farm to the laboratory. Subsequently, the tomatoes (in a PET container) were placed on the shelf in the laboratory and observed at an interval of 1, 6, and 12 days when visible spoilage symptoms such as fuzz of white, green, or grey mold and/or oozing of liquid became apparent.

### 2.3. DNA Extraction, Amplification and Illumina Sequencing

Genomic DNA (gDNA) was extracted from the selected tomatoes using Zymo DNA fungal/bacterial DNA mini prep extraction kit (ZYMO Research, Irvine, CA, USA). To begin, three samples of each tomato variety were taken on day 1 (day the tomato was harvested), day 6 (also refer to day2) and day 12 (also refer to day3), they were aseptically transferred while wearing gloves into a zip lock transparent plastic bag and subsequently subjected to grinding using a stomacher apparatus (VWR Scientific, Dublin, IE, USA) one bag for each tomato variety), and mixed properly afterwards. The gDNA was extracted according to the manufacturer’s instructions and a Nanodrop spectrophotometer (IMPLEN, nanophotometer NP80, München, Germany) was used to determine the purity and quality of the resulting total gDNA. In addition, the integrity of the gDNA was verified by visualizing it on a 1% agarose gel (*w*/*v*) stained with SYBR Safe DNA gel stain (Invitrogen Co., Carlsbad, CA, USA). Prior to further analysis, the extracted gDNA was stored at −20 °C to maintain its integrity at the time of use. 

Using polymerase chain reaction (PCR), the V4 hypervariable regions of 16S rRNA [28] and conserved fungal ITS 1 region genes [29,30] of the purified gDNA were amplified. The forward and reverse primers, 515 F and 806 R for the 16S rRNA gene and the forward and reverse primers CTTGGTCATTTAGAGGAAGTAA and GCTGCGTTCTTCATCGATGC for the ITS primers were used (www.mrdnalab.com (accessed on 11 April 2022), Shallowater, TX, USA). The barcode sequence was incorporated into both the 16S and ITS reverse primers. The V4 and ITS 1 regions of 16S rRNA and ITS1 genes, respectively, were amplified in a 96 well microtiter plates with 50 ng template DNA with a total reaction volume of 50 mL using HotStarTaq Plus Master Mix Kit (Qiagen, Germantown, MD, USA) and 96 barcoded reverse primers (www.mrdnalab.com (accessed on 11 April 2022), Shallowater, TX, USA). Cycling parameters included a 5-min initial denaturation at 95 °C, 30 cycles of 30 s at 95 °C (denaturing), 40 s at 53 °C (annealing), and 60 s at 72 °C (elongation), and a 10-min extension at 72 °C. The PTC-100 thermal controller (BIORAD, Hercules, CA, USA) was used for all the reactions. Each barcoded primer pair also had a negative control without a template. Gel electrophoresis on a 2% agarose gel stained with SYBR Safe (Invitrogen Co., Carlsbad, CA, USA) confirmed the presence of amplicons and the relative intensity of bands. The QuantiT TM PicoGreen^®^ dsDNA assay was used to determine the quality of PCR products (Life Technologies, Carlsbad, CA, USA). According to their molecular weights and DNA concentrations, the PCR amplicons were pooled together in equal amounts (100 ng). Amplification primers and reaction buffer were removed from each sample using calibrated Ampure XP beads (Beckman-Coulter, Pasadena, CA, USA). An Illumina DNA library was then made from the pooled and purified PCR products. In accordance with Illumina MiSeq (2 × 250-bp) specifications, sequencing was carried out at MR DNA (www.mrdnalab.com (accessed on 3 May 2022), Shallowater, TX, USA).

### 2.4. Sequence Analysis

Sequences were binned by samples using the specific barcode sequences and trimmed by removal of the barcode and primer sequences using MR DNA Software Package “FASTqProcessor” v.20.01.14 (http://www.mrdnalab.com/mrdnafreesoftware/fastqprocessor.html (accessed on 20 July 2022)). Reads quality were evaluated using the Divisive Amplicon Denoising Algorithm 2 (DADA2) pipeline v.1.18 (https://iallali.github.io/DADA2 pipeline/16SrRNA DADA2 pipeline.html (accessed on 20 July 2022)), a bioinformatics pipeline that retrieves biological sequences from reads by modeling the Illumina-sequencing errors [31]. In summary, the core DADA2 algorithm was followed by filtering raw sequence reads, inferring amplicon sequence variants (ASVs), and merging of pairs; construction of ASVs table, and removal of identified chimeras. Reads with a minimum length of 220 bp for reverse reads and a maximum length of 230 bp for forward reads were trimmed, a criterion intended to eliminate ambiguous base calls. This selection process was undertaken with the aim of estimating the error rate, as elucidated in reference [32].

Similar to the bacterial pipeline, the fungal ITS version of the DADA2 pipeline v. 1.8 (http://benjjneb.github.io/dada2/ITS workflow.html (accessed on 20 July 2022)) used the same workflow. The ASVs obtained were taxonomically classified using SILVER and the UNITE 8.3 databases for bacteria and fungi, respectively, and diversity analyses were inferred in R Studio (Phyloseq package) [31,33]. The “phyloseq” package [34] was used to transform all the ASV tables that the DADA2 pipeline generated into phyloseq objects v.1.24.2. The ASVs were classified as “Exact” (perfect match to a true sequence), and chloroplast was removed.

### 2.5. Statistical Analysis

The “estimate richness” function in “phyloseq” was used to determine the alpha diversity index and significance differences in alpha diversity was calculated using a Kruskal–Wallis test. Bray–Curtis distance matrices were constructed using the R package ‘vegdist’ (‘vegan’) to determine the differences between the two cultivars. Using metric multidimensional scaling (MDS), the dissimilarities between the cultivars were evaluated based on the sequence data. Using Kruskal’s stress, the MDS, which is based on the Bray–Curtis distance matrix, was used to evaluate how well the generated ordination fits the provided dissimilarities. According to Clarke [35], Kruskal’s stress values less than 0.2 represent plots with good ordination, and PERMANOVA [36] was used to determine the significance differences in beta diversity. In addition, analysis of similarity (ANOSIM) [37] to identify the differences in bacterial and fungal composition between the tomato cultivars based on the presence/absence of differences according to the Bray–Curtis distance matrix was also performed. The R-value generated by the ANOSIM with 719 permutations ranges from 0 (totally similar) to 1 (completely dissimilar) [38]. If not specified, *p*-value > 0.05 was considered statistically insignificant. The relationship between the cultivar was further confirmed with a heatmap which was generated using an online web tool ClustVis (https://biit.cs.ut.ee/clustvis/ (accessed on 25 April 2023)). A bar chart was used to visualize the relative abundance across all taxa.

## 3. Results

### 3.1. Characteristics of the Study Population, Microbial Taxa and Sequence Depth

The population characteristics of the bacteria and fungi population (structure) of investigated jam and round tomatoes across storage time are depicted in Table 1 while the population counts are shown in Figure 1A, B, respectively. The taxonomic assignment of the sequences revealed a total of 472 bacterial taxa from a total high-quality sequence reads of 4,467,572 with average length of 254 bp while 256 fungal taxa from a total high-quality sequence reads of 769,480 with average length of 240 bp in the tomato microbiota of the examined cultivars (Table 1). After quality filtering and removal of chimeric sequences, the obtained quality sequence reads for bacteria on day 1 (referred to as Jam Tomato 1 and Round Tomato 1) were 612,139 and 761,957, 504,872 and 1,044,763 on day 6, and 507,943 and 1,035,898 on day 12. Additionally, the sequence reads for fungi are 96,662 and 142,268 (for day 1), 142,534 and 185,710 on day 6 while 54,464 and 147,842 was depicted on day 12 (Table 1). It was observed that the number of bacterial sequence reads was greater than that of fungal sequence reads, and round tomato had high sequence reads during all storage periods for both bacteria and fungi. 

The rarefaction of the sequences at the sequence depth of 40,000 was sufficient (coverage was greater than 99 percent) to compare microbial diversity between cultivars and storage periods; showing that all samples reached saturation phase (Appendix A). The relative abundance of the organisms in relation to the studied parameters (cultivars and storage periods) as depicted by the Venn diagram (Figure 2) revealed that 79 ASVs were common to the two cultivars for the bacterial communities (Figure 2A), while 46 were common for fungal communities (Figure 2B). Round tomato was observed to yield a greater number of ASVs for both bacterial and fungal communities as compared to jam tomato (Figure 2). 

### 3.2. Microbial Biodiversity of the Studied Tomato Cultivars (with Respect to Storage Period) 

#### 3.2.1. Bacterial Alpha Diversity

The results of bacterial richness and diversity between the two studied tomato cultivars as a function of three alpha diversity indices as shown in Figure 3. The observed alpha diversity index (Figure 3A) revealed that the species of jam tomato had the highest richness and abundance (number of taxa) on day 12 and the lowest abundance on day 6 (Figure 3A). While in round tomato the highest levels of richness and abundance were observed on day 1, this decreases drastically on day 6 and then increases slightly on day 12 (Figure 3A). Shannon alpha diversity, which measures distribution (evenness), showed higher diversity in day 12 jam tomato and day 1 round tomato followed by jam tomato day 6 and 1 and round tomato at day 6 and 12 (Figure 3B). The round’s diversity and richness decrease while the jam’s increase with storage time. Tomato fruits observed for bacteria on day 1 had Simpson’s diversity indices of 0.49 and 0.65, for jam and round tomato, respectively, while those observed on the 6th and 12th day had (0.72 and 0.08) and (0.85 and 0.12), respectively (Figure 3C). After 12 days of storage, none of these indices demonstrated significant differences in diversity richness between cultivars (*p*-values greater than 0.05).

#### 3.2.2. Fungi Alpha Diversity

As with the bacteria, while the highest fungal richness and abundance (observed alpha diversity) was found with round tomato at day 1, it was also observed that there was a decrease trend in species abundance with prolong storage periods (for both jam and round) (Figure 4A). Likewise, the Shannon and Simpson alpha diversity indices for both tomato cultivars revealed greater diversity and evenness on day 1 (Figure 4B,C). The tomato fruits assessed on days 1, 6, and 12 had a Simpson’s diversity index of (0.79, 0.34, 0.39) for jam tomato and (0.88, 0.80, 0.44) for round tomato, respectively. In contrast to that of bacteria, round tomato at day 12 had the least even distribution, followed by jam tomato at day 6. Similar to the results for bacterial alpha diversity, the results for fungi alpha diversity revealed greater diversity and richness in round tomato compared to jam tomato, although both decreased in diversity and richness as storage period increases. However, none of these indices demonstrated significant differences across storage periods and within the cultivars with *p*-values greater than 0.05.

#### 3.2.3. Beta Diversity

The “Bray–Curtis dissimilarity metrics” based on MDS revealed a variation of 72.6% between the x (43.3%)- and y (29.3%)-axes in terms of diversity between cultivars. According to the results of the MDS plot (for bacteria), jam and round tomato on day 1 of storage period clustered together (Figure 5A). However, on the 6th and 12th days of storage, jam and round tomato clearly differentiated on the MDS plot (Figure 5A). While the fungi MDS plot showed a variation of 46.2% (*x*-axis (24.3%) and *y*-axis (21.9%)), no clustering between the two cultivars regardless of storage duration was observed as they are all scattered on the plot (Figure 5B). The ANOSIM based on the Bray–Curtis dissimilarity index revealed no significant difference between the bacterial and fungal community structures of the two tomato cultivars [(R = 0.2222, 0.2222, respectively); (*p* = 0.3; *p* = 0.3), respectively]. The PERMANOVA test on Bray–Curtis’ dissimilarity plot revealed no significant difference in the bacterial R^2^ = 0.26731, *p* = 0.3 and fungal R^2^ = 0.2272, *p* = 0.2 communities between the cultivars. The relationship between cultivar and storage period in terms of microbial abundance and/or compositions was further established as shown in a heatmap (Appendix A). 

### 3.3. Taxonomic Abundance in the Jam and Round Tomato Fruit Cultivars 

#### 3.3.1. Bacteria

Most of the microbiome across the investigated storage period (>1% abundance) was dominated by Proteobacteria (83.83%) at the phyla level (Appendix A). At the class level, 72.5% were Gammaproteobacteria with Clostridia, and Bacteroidia accounting for 1.5% of the total abundance (Appendix A). Enterobacterales made up 62.83% at the taxonomic order level while *Erwiniaceae* (34.5%) (Appendix A), *Enterobacteriaceae* (23.3%) and *Lactobacillaceae* (12.5%) are observed at high abundance at family level (Appendix A). At the genus level, *Pantoea* (28.67%) and *Klebsiella* (25%) predominantly dominated (Appendix A).

The ASV-based taxonomic classification for bacteria revealed 1 kingdom, 8 Phyla, 11 Classes, 40 Orders, 77 Families, and 131 Genera. Both cultivars were characterized by more than 90% of the phylum Proteobacteria on day 1, but as the storage period lengthens, Proteobacteria decreases in jam and increases in round tomato (Figure 6A). This is followed by Firmicutes, which prevailed in JamTomato2 with relative abundances greater than 60% (Appendix A). Actinobacteriota abundance increases in jam tomato with increasing storage period while decrease was observed for round tomato (Appendix A). Gemmatimonadota, Acidobacteriota, and Deinococcota are only present in RoundTomato1 (Appendix A).

Gammaproteobacteria constitutes over 70% of bacteria at class level, although a reduction was observed in jam tomato from 93–77% (from day 1–12) while an increase was observed in round tomato from 40–99% (day 1–12), Similarly, in jam tomato, the bacteria class, Bacilli, increased from 2% (day 1) to 69% (day 6) before experiencing a significant decline to 7% (day 12). However, in round tomato, only 1% was observed on day 1. Additionally, the class, Alphaproteobacteria decreases from 6% (day 6) to 4% (on day 12) for jam tomato, whereas for round tomato, it dropped from 27% (day 1) to 1% (day 12). Noticeably, the bacteria classes, Clostridia and Bacteriodiota are identified in jam tomato at high abundance compared to round tomato (Appendix A). 

The Enterobacteriales are found in substantial quantity in both cultivars particularly in the round tomato, where they increased from 30% to 99%, as against jam tomato, where they decreased from 73% to 49%. The difference in abundance was noted in both cultivars. In a similar vein, a significant proportion of Lactobacillales is witnessed on day 6 (69%) in jam tomato which moderately decreased to 7% on day 12 though at day 1, it was very low (1%). This bacteria at this order level were only detected in round tomato at abundance (<1%) throughout the storage intervals. In addition, there was an increase in the abundance of Pseudomonadales in jam tomato from 17% on day 1 to 25% on day 12 while there was a decrease in round tomato from 7% to 0.07%, respectively. Jam tomatoes have bacterial orders such as Lachnospirales, Flavobacteria, Acetobacterales, Bacteriodales, Oscillospirales, and Cardiobacteriales at abundance greater than 1% compared to round tomato. Whereas round tomatoes contain the bacterial order Rickettsiales at abundance greater than 1% compared to jam tomato (Appendix A).

Eight (*Erwiniaceae*, *Enterobacteriaceae*, *Lactobacillaceae*, *Pectobacteriaceae*, *Pseudomonadaceae*, *Moraxellaceae*, *Acetobacteraceae,* and *Lachnospiraceae*) out of the 78 families (Appendix A) identified in the ASV-based taxonomic diversity of bacteria in jam and round tomato had a higher relative abundance (>1% abundance across storage periods) whereas the others had a lower relative abundance (<1%). *Erwiniaceae* exhibited a significant rise in round tomato from 1% (day 1) through 98% (day 6) and down to 97% (day 12) as compared to only 9% in jam tomato on day 12. In contrast to the afore mentioned, a greater proportion of *Enterobacteriaceae* was seen in jam tomato (72%) on day 1 and 34% (day 12) compared to round tomato (29% and 2%, respectively). Comparatively, jam tomatoes similarly reflected a greater proportion of *Moraxellaceae*, *Lachnospiraceae*, *Acetobacteraceae*, *Lactobacillaceae*, *Pectobacteriaceae*, and *Pseudomonaceae* than round tomato (Appendix A). 

Twenty-five of the 131 identified Genera (Appendix A) exhibited a high proportion of significant abundance during the storage period in both cultivars. The proportion of *Pantoea* from 0% to 90% to 82%, was notably high in round tomato particularly on days 6 and 12; however, this genus is absent in jam tomato while a significant percentage of *Kosakonia* (35%) and *Pseudomonas* (29%) was observed on day 12 of jam tomato (Figure 7A). On day 1 of both cultivars, the genus *Klebsiella* is observed at a high percentage (above 70%). *Leuconostoc* (28%, 4%), *Pseudomonas* (31%, 7%), *Lonsdalea* (29%, 8%), *Weissella* (43%, 1%), and *Acinetobacter* (18%, 15%) are prevalent in both jam and round cultivars, albeit in varying proportions throughout the storage period while *Bacillus* (0%, 1.01%), *Stenotrophomonas* (0%, 4%), *Serratia* (1%, 0%), *Acetobacter* (10%, 0%), *Liquorilactobacillus* (4%, 0%), *Allorizobium* (0%, 1%), *Dysgonomonas* (4%, 0%), *Kurthia* (1%, 0%), *Lactococcos* (1%, 0%), and *Achromobacter* (0%, 1%) are present in low abundance (Appendix A).

#### 3.3.2. Fungal Taxonomic Abundance in the Jam and Round Tomato Fruit Cultivars

The fungal diversity taxonomic ASV-based classification revealed 1 kingdom, 3 Phyla, 14 Classes, 29 Orders, 49 Families, and 61 Genera.

On the other hand, the mycobiome abundance for the tomato across the storage period (abundance > 1%) were dominated by Ascomycota (97.17%) followed by Basidiomycota (2.83%) at the phyla level (Appendix A). At the class level were Saccharomycetes (41.67%), Dothideomycetes (37%), and Sordariomycetes (15%) observed to be the most abundant (Appendix A). Saccharomycetales (41.67%) and Pleosporales (26.83%) were the most abundant at the taxonomic level of order (Appendix A). While at the family level, *Pleosporaceae* (26.83%) and *Saccharomycodaceae* (22.83%) were more dominated (Appendix A); however, *Hanseniaspora* (23.17%), *Stemphylium* (14.5%), *Alternaria* (13.3%), and *Cladosporium* (9.7%) were observed to exhibit the major abundance of the fungi community at the genus level with the least abundance being *Trichosporon* (1.7%) (Appendix A). 

The fungi revealed Ascomycota constituting more than ninety percent of all fungi communities, out of the three identified fungi phyla (Figure 6B). For instance, 91% abundance of the Ascomycota are found in jam tomato day 1, which rises to 100% in both jam tomato day 6 and 12 The trend is the similar for round tomatoes, except on day 6 where 99 percent abundance was obtained. Ascomycota was then followed by the Basidiomycota accounting for 9% and 6% abundances on day 1 of jam and round tomato, respectively. 

The class, Dothideomycetes increases from 52% and 76% in round tomato across storage periods and between 1% and 75% in jam tomatoes (day 1 and 12, respectively). Saccharomycetes showed a strong increase from day 1 to day 6 and then declined on day 12 in jam (38, 98, and 11%, respectively) as well as in round tomato (15, 86, and 2%, respectively); however, the decrease for round tomato is more pronounced. A decrease is observed in Tremellomycetes and Eurotiomycetes in both cultivars while Sordariomycetes increase in round tomato (Appendix A).

Both cultivars (jam and round) at the order level showed a rise in Saccharomycetales from day 1 (38, 15%, respectively) and day 6 (98, 86%, respectively), followed by a decline on day 12 (11, 2%, respectively). Meanwhile, on day 12, a higher abundance of Pleosporales (79%) for jam tomato and 75% round tomato was detected. Similarly, Trichosporanales (9%) and Diaporthales order (48%) are mostly observed in jam tomato, while the Capnodiales (57%), Eurotiales (19%), and Hypocreales (29%) are observed in round tomato (Appendix A) compared to others.

Fungi communities at the family level revealed a substantial percentage of *Pleosporaceae* on day 12 in both jam (79%) and round (75%) (Appendix A). Similarly, *Saccharomycodaceae* was discovered in jam (82%) and round tomato (46%), respectively, on day 6. Notable prevalence or abundance of *Cladosporiaceae* (57%), *Debaryomycetaceae* (40%), *Nectriaceae* (23%), and *Aspergillaceae* (18%), in round tomato, as well as *Diaporthaceae* (48%), *Trichosporonaceae* (9%), *Dipodascaceae* (10%), and *Pichiaceae* (15%) in jam tomato are observed (Appendix A).

Among the fungi genera with abundance greater than 1, *Hanseniaspora* exhibited the highest percentage across the entire storage period in both jam and round tomatoes. Notably, on day 6, *Hanseniaspora* was particularly prominent in both cultivars, accounting for 84% in jam tomato and 46% in round tomato. However, its abundance decreased significantly on day 12 to only 0.45% in jam tomato and 0.93% in round tomato (Figure 7B). In terms of the second most abundant genus, *Stemphylium* displayed high abundance on day 12 specifically in jam tomato, representing 85% of the total genera. In contrast, its abundance in round tomato on day 12 was relatively lower, accounting for only 1%. Furthermore, when comparing jam and round tomatoes on day 12, we observed that *Alternaria* and *Fusarium* had higher abundance in round tomato, with percentages of 74% and 23%, respectively. In contrast, jam tomato had relatively lower percentages of these genera, ranging from 2% to 10%. The remaining genera occurred at low abundance on day 12 in both jam and round tomatoes (Appendix A).

## 4. Discussion

Fruits and vegetables contain a large variety of bacterial and fungal taxa, which vary considerably by fruit types [39]. Moreover, certain fungal taxa may metabolize organic acids present in tomato, thereby raising the pH to stimulate the proliferation of bacterial pathogens [40]. Hence, knowing tomato’s microbial diversity is the first step in controlling its microbial growth, extending shelf life, and enhancing its quality and safety [40,41]. Both 16S rRNA and ITS genes have been reported to improve the knowledge of microbial diversity in food and environmental samples by the characterization of complex microbial communities [42]. Diverse fruit morphologies, chemical compositions, and metabolic activity provide each fruit a distinct ecological niche, favouring particular microbial communities [40]. These variables may account for the differences in biodiversity found between the two cultivars throughout different storage periods in this study. The relative abundance, diversity, and composition of microbial communities were found to be affected by cultivar type and storage period in this study. Thus, confirming that the bacteria and fungi communities present in tomato may be a consequence of the cultivar type [13] and the storage period [43]. The high sequence reads across the storage periods for round tomato as compared to jam tomato may be due to the high-water activity of round tomatoes, which creates a favourable environment for the formation of diverse microbial communities [44]. This also may indicate the reason why bacteria abundance is high in this study as this environment may be said to favour bacterial growth [44]. Bacterial communities (4467572 sequence reads) were five times more abundant than fungal communities (769480 sequence reads), indicating the bacterial community may be essential for post-harvest management measures, such as determining the best type of biological control in reducing postharvest spoilage and consequently extend tomato shelf life.

The alpha diversity indices (Observed, Shannon, and Simpson) of tomato microbiome revealed the species relationship within the cultivars. The high diversity and richness in both jam and round tomato on day 1 may be attributable to the nutritive content of tomato, which creates a favourable environment for the formation of different microbial communities [44]. In addition, it has been reported that a healthier environment could produce a more stable and diverse microbial ecosystem [45,46]. An increase in bacterial abundance in jam tomato may be the result of a shift from slightly acidic to alkaline pH of stored tomato, which promotes bacterial growth [44,47]. ANOSIM and PERMANOVA revealed that the beta diversity of bacterial organisms inhabiting jam (0.3) and round (0.3) tomato fruits did not differ significantly between the two cultivars. However, the duration of storage affects the bacterial structure of tomato ecosystems. On day 1, bacteria clustering between cultivars (jam and round) was confirmed. Similarly, the fungi ANOSIM and PERMANOVA analyses revealed no statistically significant differences (0.3 and 0.2, respectively) between cultivars, despite the absence of a distinct clustering of fungi. The results of this investigation confirmed that, the bacterial community structure of jam and round tomatoes on day 1 is comparable, and that the time of storage has a significant impact on the bacterial populations of tomato cultivars while storage period does not have effect on fungal communities. In addition to the fact that this observation is consistent with the result of alpha diversity, it also agrees with the work of Zhimo, et al. [48], who reported an increase in the bacterial and fungal diversity among three cultivars of apples during fruit development and storage with no significant difference among the cultivars, even though our findings revealed that fungal diversity decreased.

The 16S rRNA and the ITS gene markers are currently the most important targets when determining the diversity and quantification of the relative abundance of taxa at various levels of bacterial and fungal in a given environment [49]. Considering the structure of the microbial communities, the microbial composition of both cultivars on day 1 is comparable, but their relative abundance varies. This may be due to the fact that the cultivars are cultivated on the same field and more importantly each microbial community is distinct to their specific environments [50]. Whereas the differences in microbial communities as the storage period lengthens may be a result of the fruit morphologies, chemical compositions, and metabolic activity provided by each cultivar [40]. These factors provide each fruit a distinct ecological niche, favouring specific microbial communities and as the day of storage period extends, the microbial composition differs. Overall, bacterial diversity was greater than fungal diversity, with 472 and 256 taxa, respectively. The effect of storage period on microbial taxonomic abundance was mostly represented by a decrease in microbial diversity, which led to the dominance of the phylum Proteobacteria. Although, it has been hypothesized that facultative anaerobic Proteobacteria contribute to the homeostasis and stability of the anaerobic environment of the human gastrointestinal tract, they play a crucial role in preparing the gut for colonization by the stringent anaerobes essential for optimal gut function [51]. However, nonetheless, an excess of Proteobacteria has resulted in dysbiosis of the gut microbiota [52]. The recent literature confirms that the bacterial phyla Firmicutes and Bacteroidetes, and to a lesser extent Proteobacteria, dominate the gut microbiota of healthy individuals [52]. This investigation revealed that jam tomato contained more Firmicutes than round tomato, particularly on day 6, while Bacteroidetes decreased in both cultivars. The phylum Firmicutes is well known for absorbing calories from food, resulting in weight gain [53]. Although round tomato showed a lesser amount of Firmicutes, it is more diverse (presence of Gemmatimonadota, Acidobacteriota, and Deinococcota) than jam tomato thus, may be said to be healthier than jam tomato. For the fungi at relative abundance greater than 1, Ascomycota and Basidiomycota predominated. The high prevalence of Ascomycota in both cultivars may be attributed to Ascomycota’s non-fastidious nature, which enables it to thrive in tough environments with low nutrition levels [54]. Although Ascomycota have been found to be plant- and insect-pathogenic fungi [55] and corroborated with findings from a variety of plants and fruits, including grapes, cherries and pear [56,57,58,59]. Thus, may justify why it predominates over other fungi phylum as observed in this study. This result is consistent with the report of previous study where Ascomycota and Basidiomycota were most represented and most abundant phyla in minimally processed fruits and vegetables [21].

The high abundance of Gammaproteobacteria, Alphaproteobacteria, and Dothideomycetes, Saccharomycetes, and Saccharomycetes, as well as Sordariomycetes suggest that these classes of bacteria and fungi as major microorganisms responsible for fruit spoilage [60] and have been reported as spoilage and pathogenic organisms of both fruits and humans [60]. The prevalence of Gammaproteobacteria, particularly the order Enterobacteriales, the family *Erwiniaceae*, and the genus *Pantoea*, is conspicuous within the high abundance found in round tomato. Despite its versatility, *Pantoea* is known to manifest as either a commensal or a pathogenic organism [61], with reports of its isolation from plants and soil, as well as instances of infecting pepper fruits [62]. Hence, the notable presence of *Pantoea* in high abundance on days 6 and 12 in round tomato could potentially suggest a pathogenic role. Conversely, in the context of jam tomato, the prevalence of class Bacilli, order Bacillales, family *Bacillaceae*, and genus *Bacillus* is distinctive. The class Bacilli has been associated with spoilage of fruits and vegetables [63] although the presence of this class including Bacteroidetes has also been attributed to the improvement of plant growth and health [64,65]. In the realm of fungi, the microbial profile of jam tomato showcases a significant prevalence of Saccharomycetes, order Saccharomycetales, family *Saccharomycodaceae*, and genus *Hanseniaspora*. Meanwhile, round tomato demonstrates elevated abundance in the class Dothideomycetes, order Pleosporales, family *Pleosporaceae*, and genus *Alternaria*.

An intriguing observation is the proportional increase in spoilage organisms with the extension of the storage period in round tomato, encompassing taxa such as *Pantoea*, *Stemphylum*, and *Cladosporium*. Conversely, jam tomato hosts taxa such as *Hanseniaspora*, *Pichia*, and *Kasakonia*, which have been associated with robust biocontrol activity against fungal pathogens. 

Notably, most of the core bacterial genera such as *Acinetobacter*, *Klebsiella*, *Weissella*, *Pantoea*, *Kosakonia*, *Lonsdalea*, *Pseudomonas*, and *Leuconostoc* identified by the culture-independent technique are not the most frequently isolated spoilage organisms when using the culture-dependent method. *Bacillus*, *Klebsiella*, and *Staphylococcus*, which were frequently reported as tomato spoilage organisms using culture-dependent method [13], are found in low abundance though, *Klebsiella* is observed at high abundance on day 1. The significant quantity of *Klebsiella* in both cultivars on the day of harvest may have resulted from human excrement contaminating the soil [66]. 

The high abundance of Leuconostoc, Lonsdalea, Pseudomonas, Kosakonia, Weissella, Acinetobacter, Stenotrophomonas, Myroides, Acetobacter, Liquorilactobacillus, and Dysgonomonas in both cultivars at day 6 and day 12 at varying percentages suggests that cultivar host–bacterium interactions may be driven by intrinsic factors and may result in variations in the abundance of some taxa [67]. In addition, the existence of these microorganisms on days 6 and 12 suggests they may probably be spoilage or harmful bacteria. However, Kosakonia and Pseudomonas were previously identified from maize, wheat, and rice and are known to promote plant growth due to their ability to fix atmospheric nitrogen and secrete plant growth hormone [68]. 

The presence of the fungal taxa belonging to “*Alternaria*, *Fusarium*, *Meyerozyma*, *Pichia*, *Candida*, *Hanseniaspora*, *Debaryomyces*, *Cladosporium*, *Diaporthe*, *Stemphylium*, and *Penicillium*” found in both cultivar at day of harvest could indicate that they are the core fungal taxa of tomato. This may indicate that they are microorganisms that initially inhabited the tomato fruit, as observed by the analysis of the fungal genera. The taxa comprised yeasts (such as *Candida*, *Hanseniaspora*, *Pichia*) with strong biocontrol activity against various microorganisms and inhibitors of potential pathogenic microorganisms [69]. *Hanseniaspora* and *Pichia*, for instance, serve as potential biocontrol agents for tomato [70] and grape [71], respectively. While other core taxa, including *Alternaria*, *Stemphylium*, *Cladosporium*, *Diaporthe*, and *Penicillium*, are potential pathogens of pear and strawberry fruits [72,73]. Surprisingly, some of these core taxa (including *Alternaria*, *Stemphylium*, *Cladosporium*, *Diaporthe*, and *Penicillium*) are found on days 6 and 12 (and belonged to the phyla, Ascomycota), thus, might suggest them as spoilage organisms of tomato. Meanwhile, the literature reports have established that these organisms can be found on the surface of fruits, responsible for the likely rotting and toxins production that pose a health risk [74,75,76]. While the aforementioned may suggest that these predominant fungi could be responsible for tomato spoilage, it could also buttress how adaptable these organisms are and their ability to survive in a variety of environments [77]. This is in agreement with previous research, for instance, *Cladosporium* and *Alternaria* reported to be responsible for rotting and plant disease in fruits and vegetables [78]. In addition to their role in food spoilage, many microbes of these genera are sources of allergic disorders or sensitizations [21], which constitute a health risk.

Above all, findings from this study have demonstrated that consuming spoiled tomatoes, known as “utamatisi owonakele” in Zulu (South Africa) and ‘esa’ in Yoruba (Nigeria), could have detrimental effects on humans. This is because the prevalence of these microorganisms on the crop indicates the likelihood that they contribute to the microbiome composition of people who consume them.

## 5. Conclusions

This is the first longitudinal study to characterize the microbial composition of two commonly consumed tomato varieties (jam and round) in South Africa using targeted high throughput sequencing. The study found that the microbial composition of the cultivars is comparable on the day of harvest but becomes unstable and varies as storage time increases. The study established bacteria as the major spoilage organisms of tomato as compared to fungi reported in the literature. It was also noted that the microbial structure in round tomatoes was more diverse than that of jam tomatoes, despite the insignificant difference between them. Thus, validating the effect of cultivar types on tomato’s microbial composition. Overall, while the study contributes to a greater understanding of the distinct bacterial and fungal diversity profiles in jam and round tomato cultivars by highlighting species whose presence and abundance vary between the two cultivars and storage periods, further exploration of the microbial composition of tomato to the species level using long read sequencing platforms such as PacBio may be beneficial. Importantly, the identification of dominant microbial taxa at each taxonomic level during storage in this study can help formulate natural preservatives, produce ideal storage conditions, improve tomato quality and availability for consumers, reduce food waste and economic viability. More specifically, findings from this study may influence tomato cultivation policies on prioritizing post-harvest management and sustainable preservation in South Africa. 

## Figures and Tables

**Figure 1 microorganisms-11-02337-f001:**
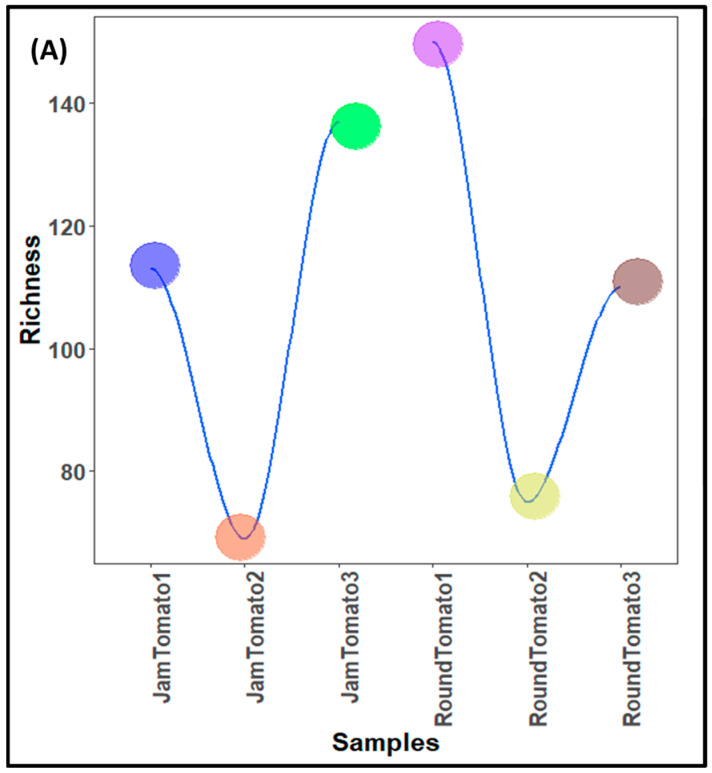
Population counts of (**A**) bacteria (**B**) fungi across storage period for jam and round tomatoes. JamTomato1 and RoundTomato1: day 1, JamTomato2 and RoundTomato2: day 6 and JamTomato3 and RoundTomato3: day 12.

**Figure 2 microorganisms-11-02337-f002:**
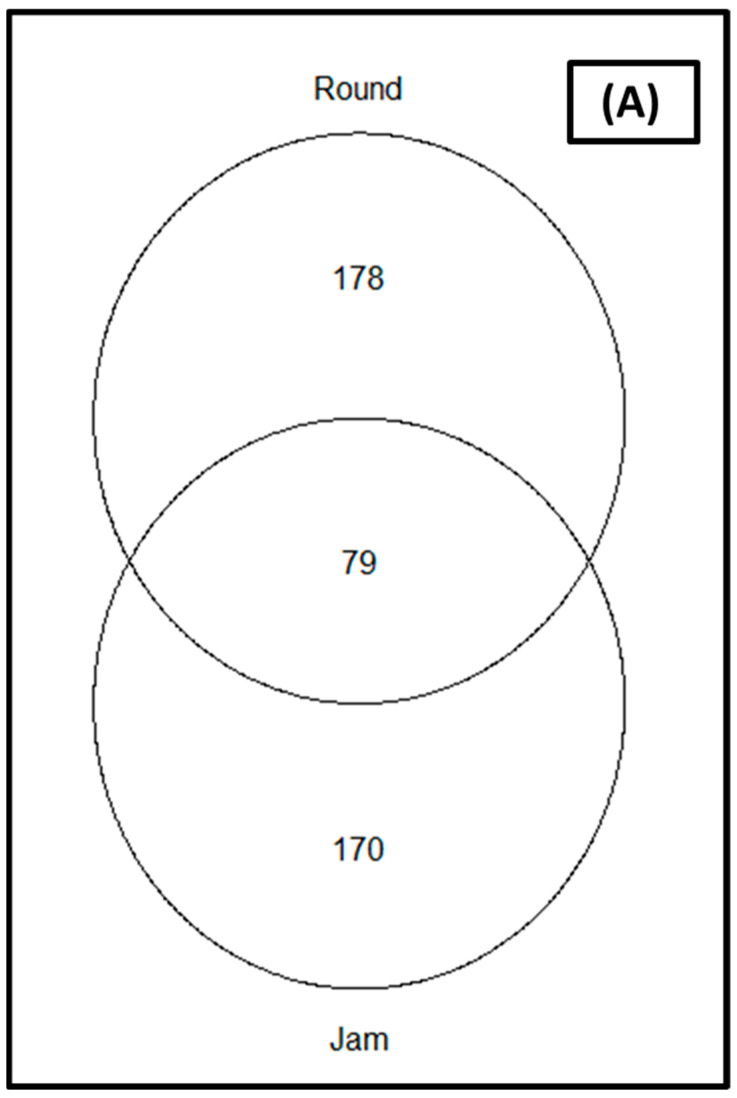
Venn diagrams showing the amplicon sequence variants (ASVs) shared between jam and round tomato for the (**A**) bacterial and (**B**) fungal communities. Only taxa with a relative abundance >1% in each cultivar are included.

**Figure 3 microorganisms-11-02337-f003:**
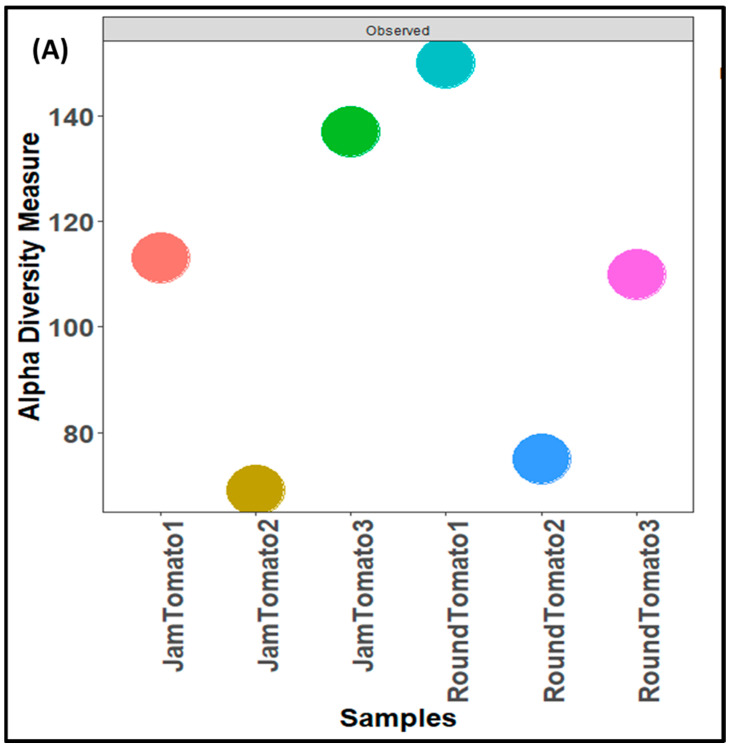
Comparison of bacterial community diversity of jam and round tomatoes over a 12 day storage period. Alpha diversity was measured by (**A**), observed (**B**), Shannon (**C**), and Simpson diversity indices. JamTomato1 and RoundTomato1: day 1, JamTomato2 and RoundTomato2: day 6 and JamTomato3 and RoundTomato3: day 12.

**Figure 4 microorganisms-11-02337-f004:**
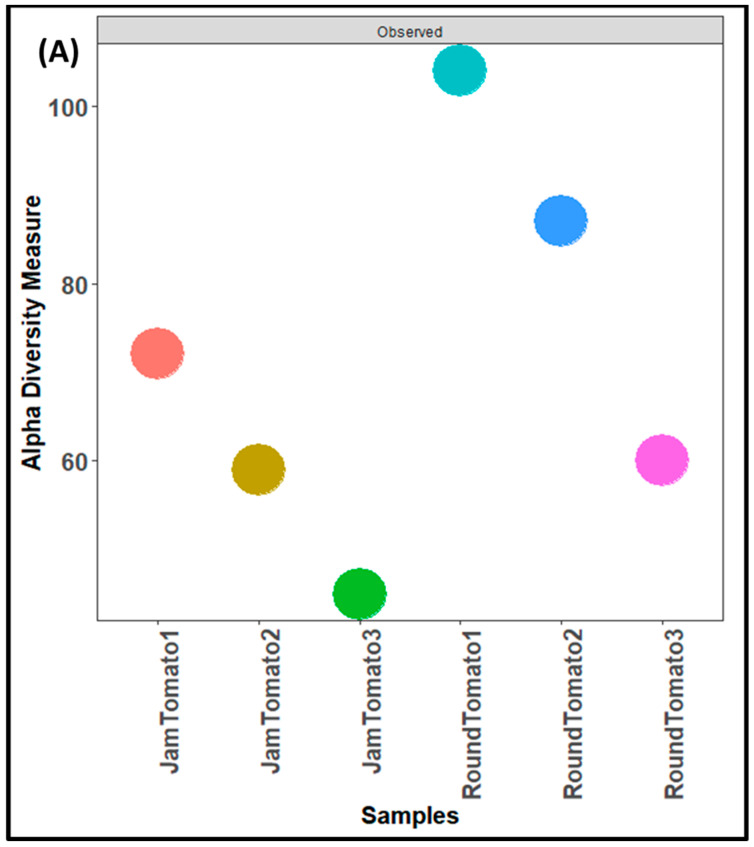
Comparison of fungal community diversity of jam and round tomatoes over a 12-day storage period. Alpha diversity was measured by (**A**) observed, (**B**) Shannon, (**C**) and Simpson diversity indices. JamTomato1 and RoundTomato1: day 1, JamTomato2 and RoundTomato2: day 6 and JamTomato3 and RoundTomato3: day 12.

**Figure 5 microorganisms-11-02337-f005:**
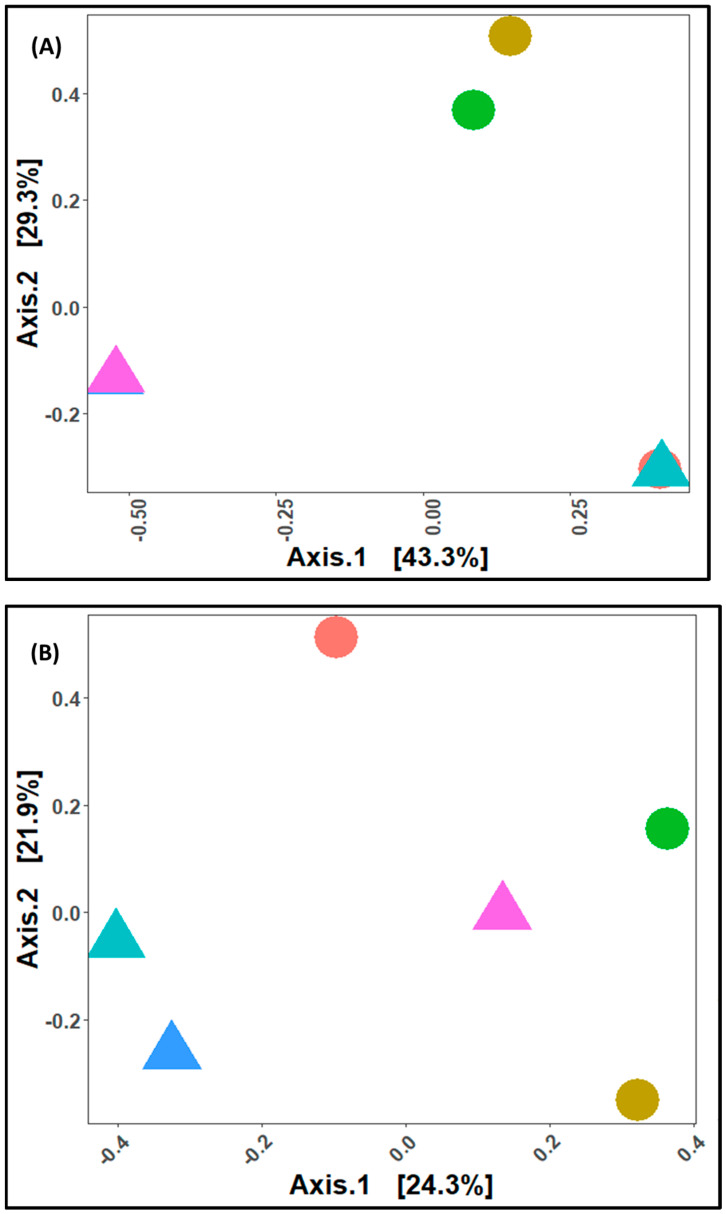
Multidimensional scaling (MDS) plot of Bray-Curtis’ distance matrix of jam and round tomato for (**A**) bacteria and (**B**) Fungi. Each shape represents the samples while colours represent the number of days. Samples with similar microbial composition tended to be in the same area of the graph, while shapes far apart from each other and represent samples with dissimilar microbiota.

**Figure 6 microorganisms-11-02337-f006:**
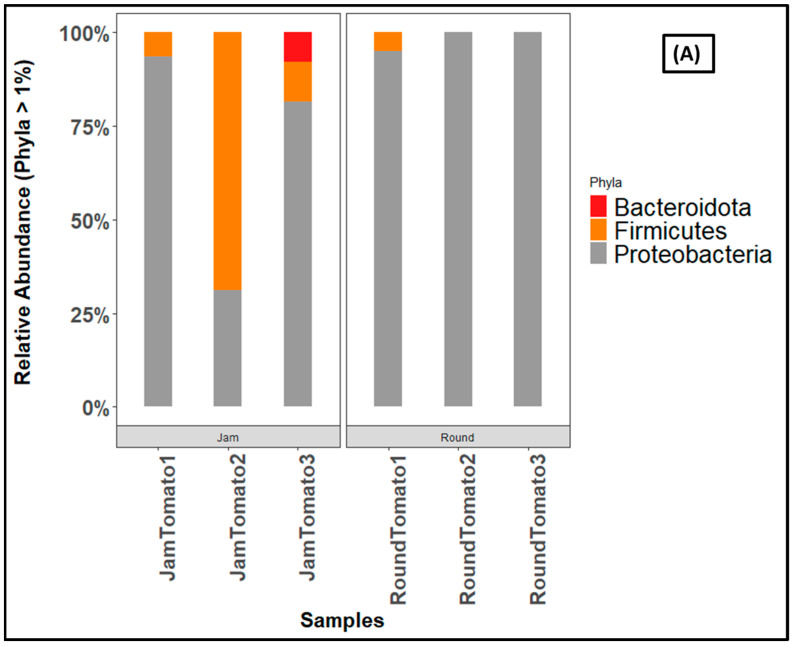
Bar charts showing the relative abundance of (**A**) bacteria, and (**B**) fungi at phylum level across storage periods. JamTomato1 and RoundTomato1: day 1, JamTomato2 and RoundTomato2: day 6 and JamTomato3 and RoundTomato3: day 12.

**Figure 7 microorganisms-11-02337-f007:**
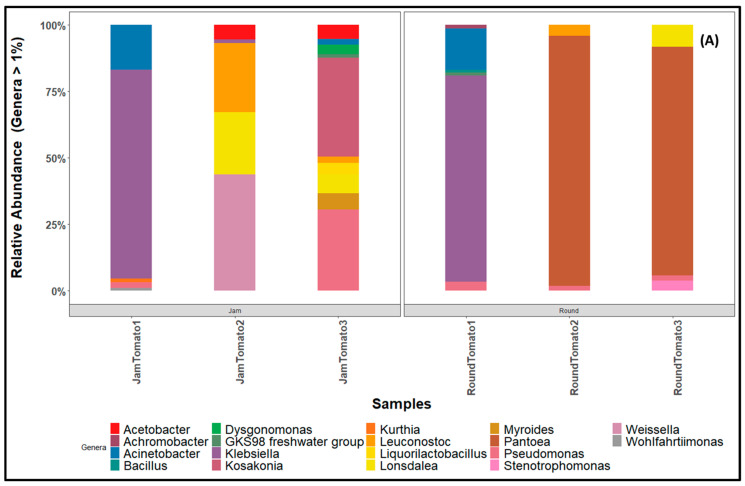
Bar charts revealing the relative abundance of (**A**) bacteria (**B**) fungi community structure at genus level across storage period for tomato cultivars (jam and round). JamTomato1 and RoundTomato1: day 1, JamTomato2 and RoundTomato2: day 6 and JamTomato3 and RoundTomato3: day 12.

**Table 1 microorganisms-11-02337-t001:** Taxonomic assignment of sequences obtained for the investigated tomato cultivars.

	Bacteria		Fungi	
Number of taxa	472		256	
Sequence reads	4,467,572		769,480	
Average length	254		240	
Bimeric sequence	Jam	Round	Jam	Round
	612139 *	761957 *	96662 *	142268 *
	504872 ^$^	1044763 ^$^	142534 ^$^	185710 ^$^
	507943 ^#^	1035898 ^#^	54464 ^#^	147842 ^#^

*: Day 1, ^$^: Day 6, ^#^: Day 12.

## Data Availability

The sequencing data derived from the 16S rRNA and ITS genes in this study have been archived in the NCBI Sequence Read Archive (SRA) database, accessible via the BioProject ID: PRJNA1013823.

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
