# Peer review of "Characterization of Microbial Diversity of Two Tomato Cultivars through Targeted Next-Generation Sequencing 16S rRNA and ITS Techniques"

_microorganisms, 2023, doi:10.3390/microorganisms11092337_

Round 1

Reviewer 1 Report

Review on “Characterization of microbial diversity of two tomato cultivars through targeted next-generation sequencing 16S rRNA and ITS techniques” for IJMS (manuscript ID microorganisms-2576373)

In this manuscript the authors present an effort to compare the microbial communities of two tomato varieties: ‘roma’ and ‘beefsteak’ and how it changes with storage process.

Unfortunately, the Supplementary files (tables) are missing, and I cannot to check the raw data or reproduce some of the results.

Comments on Introduction section:

Large part of the Intro (L37-51) devoted to the importance of tomato as the element food production, but lacking knowledge to date of tomatoes’ microbial community or previous studies review on the topic. I strongly suggest to include the recent data about the microorganisms living in the tomato plant and set the aim of the study clearer.

The following papers may help to improve the Intro:

·        Escobar Rodríguez C, Novak J, Buchholz F, Uetz P, Bragagna L, Gumze M, Antonielli L and Mitter B (2021) The Bacterial Microbiome of the Tomato Fruit Is Highly Dependent on the Cultivation Approach and Correlates With Flavor Chemistry. Front. Plant Sci. 12:775722. doi: 10.3389/fpls.2021.775722

·        Tomato microbiome under long-term organic and conventional farming  https://doi.org/10.1002/imt2.48

·        Resendiz-Nava CN, Alonso-Onofre F, Silva-Rojas HV, Rebollar-Alviter A, Rivera-Pastrana DM, Stasiewicz MJ, Nava GM, Mercado-Silva EM. Tomato Plant Microbiota under Conventional and Organic Fertilization Regimes in a Soilless Culture System. Microorganisms. 2023; 11(7):1633. https://doi.org/10.3390/microorganisms11071633

·        Core Microbiome of Solanum Lycopersicum for Sustainable Agroecosystems (book by Anamika Chattopadhyay, G. Thiribhuvanamala)

L71: what “R 94” stands for?

L83: the refences [19, 20] should be replaced with those which related to tomato plants.

My questions about Results and Discussion:

L191: The term “demography” shouldn’t be used for microbial community.

L194: Why the taxons considering at the whole (all samples of each variety)? It’s interesting to compare the changes of microbial community structure during storage.

Captions of Figures 3, 4 and 7 are unreadable. BTW what is the point using this type of diagram? The histogram seems to be the better choice.

The Venn diagram (Figure 2) has invalid proportion. Using web service DeepVenn might help.

The Discussion section shouldn’t contain introductory part (L416-L438), please interpret your data and compare with similar studies.

L444: Comparing the abundance of microbial community by reads number is incorrect.

L448: what “Observed” index stands for?

What is the point discussing the human gut microbiota (L484-L489)?

L555: How the present study covers “detrimental effects” of “consuming spoiled tomatoes” on humans?

Conclusion section should contain the summary of the findings, not the speculation for the further research (L571-L584).

Methods section comments:

·        L118: was the surface of the fruits treated with detergent or another cleaning agent before the DNA extraction?

·        L148: please specify the sequencing platform (MiSeq has different types) and read length.

·        L151: Raw reads should be made public and mentioned in the manuscript.

Some minor corrections to the text (style and spelling):

·        L117: missing space.

Author Response

Dear Editor,

Thank you for giving us the opportunity to submit a revised draft of our manuscript titled: Characterization of microbial diversity of two tomato cultivars through targeted next-generation sequencing 16S rRNA and ITS techniques to Microorganisms. We appreciate the time and effort that you and the reviewers have dedicated to providing your valuable feedback on our manuscript. We are grateful to the reviewers for their insightful comments on our paper. We have been able to incorporate changes to address most of the suggestions provided by the reviewers. We have highlighted the changes within the manuscript. We hope you find these revisions relevant, and we are ready to make further amendments if deemed necessary.

Thank you.

Here is a point-by-point response to the reviewers’ comments and concerns.

Reviewer #1

Comment 1

In this manuscript the authors present an effort to compare the microbial communities of two tomato varieties: ‘roma’ and ‘beefsteak’ and how it changes with storage process. Unfortunately, the Supplementary files (tables) are missing, and I cannot to check the raw data or reproduce some of the results.

Response: The supplementary files were initially included with the manuscript during the submission process. However, we have re-attach them in the hope that they are now accessible and readily available. Many thanks.

Comment 2 (Introduction section):

Large part of the Intro (L37-51) devoted to the importance of tomato as the element food production, but lacking knowledge to date of tomatoes’ microbial community or previous studies review on the topic. I strongly suggest to include the recent data about the microorganisms living in the tomato plant and set the aim of the study clearer.

Response: Thank you for your insightful comment. We appreciate your feedback on the manuscript and have made the necessary changes accordingly. In the revised version of the text, we have incorporated recent data on the relationships between tomato plants and their microbiomes and further elaborated on prior studies on the microbial community associated with tomatoes (lines 74-105). These additions aim to provide a more comprehensive background information on this study and findings. Furthermore, the aim of the study is better aligned with the objectives of the research. We believe that these modifications will address the concerns you raised.

Comment 3:

L71: what “R 94” stands for?

Response: R 94 is the South African currency and stands for 94 million Rands line 73.

Comment 4:

L83: the refences [19, 20] should be replaced with those which related to tomato plants.

Response: Your observation is indeed significant. However, we maintain that references [19, 20] remain pertinent as they broadly address the application of culture-independent methods in identifying fastidious or non-cultivable microorganisms through sequence marker genes. While the references might not specifically pertain to tomato plants, they still contribute to the overarching concept of employing culture-independent techniques for microbial analysis. However, reference related to tomato have been included (line 95).

Comment 5 (Results and Discussion):

L191: The term “demography” shouldn’t be used for microbial community.

Response: We appreciate your feedback. The term "demographic" has been replaced with "population" as per your suggestion (page 5, line 214). We recognize that "demographic" is more aptly associated with human contexts, considering factors such as age, race, and sex.

Comment 6:

L194: Why the taxons considering at the whole (all samples of each variety)? It’s interesting to compare the changes of microbial community structure during storage.

Response: Many thanks for this suggestion. This observation is observed in the manuscript where Gammaproteobacteria prevail in round tomatoes while, the class Bacilli prevail in jam tomatoes indicating the dominance of one taxon over the other as a result of interactions between different microbial groups. Furthermore, the elevated abundance of Pantoea in round tomatoes on days 6 and 12, coupled with its potential pathogenic nature, suggests a potential shift in microbial dynamics towards more pathogenic strains as storage progresses. This is contrasted by jam tomatoes hosting taxa with biocontrol activity, potentially helping to mitigate the proliferation of pathogenic microorganisms Lines (Lines 531-562, 571-576).

Comment 7:

Captions of Figures 3, 4 and 7 are unreadable. BTW what is the point using this type of diagram? The histogram seems to be the better choice.

Response: We sincerely apologize for any inconvenience caused by the presentation of our original Figures 3, 4, and 7. This was not our intention, and we deeply appreciate your attention to detail. We have taken your feedback into account and have made the necessary modifications to the figures to ensure clarity and proper visibility.

Regarding the choice of using bar charts to represent our results, we acknowledge your point. Indeed, our data is categorical, and bar charts effectively convey these categorical variations, aiding in the clear visualization of our findings. We believe that the utilization of bar charts aligns well with the nature of our data and facilitates the effective communication of our results.

Comment 8: The Venn diagram (Figure 2) has invalid proportion. Using web service DeepVenn might help.

Response: We genuinely apologize for the oversight in the initial proportion representation. We have replaced the figure with an updated version that accurately depicts the proportions. Your suggestion about using a web service Deepven diagram to illustrate differences and similarities between samples is well-taken and can be explore in the future. Many thanks for your meticulous insight.

Comment 9: The Discussion section shouldn’t contain introductory part (L416-L438), please interpret your data and compare with similar studies.

Response: This observation has been amended in the manuscript lines (452-455).

Comment 10: L444: Comparing the abundance of microbial community by reads number is incorrect.

Response: We appreciate your meticulous observation and concur with your insight. However, while comparing microbial community abundance solely by read numbers might be inappropriate (due to length of sequences, sequencing depth, and biases in sequencing), in this specific case, the comparison was drawn between the high sequence reads and the water activity of round tomatoes (lines 471-479). This relationship, in turn, relates to the conducive environment fostered by the water activity which favoured bacterial community. As such, the correlation between bacterial abundance and water activity was the focus of this comparison. Although read number can provide an estimate of the relative abundance of different microbial taxa in a sample.

Comment 11: L448: what “Observed” index stands for?

Response: Thank you for your question. The "Observed" alpha diversity index serves as a measure indicating the number of distinct species within each sample, a metric commonly referred to as "richness" (251-255).

Comment 12: What is the point discussing the human gut microbiota (L484-L489)?

Response: We greatly appreciate the insightful question raised by the reviewer. This study focuses on a comparative analysis of the microbial communities present in jam and round tomato varieties during storage. While this study focuses on microbial diversity in tomatoes, we acknowledge the reviewer’s point regarding the potential relevance to human gut microbiota. While this study does not directly address the human gut microbiota, it is important highlighting that the types of microorganisms within our diet, such as those found in tomatoes, can have indirect effects on human health. This has lent credence to the health benefit of tomato. Numerous studies have established a strong connection between dietary habits and gut microbiota composition, which can impact overall health and well-being. Their diverse microbial communities, as uncovered by this research, might have implications for the nutritional and health benefits associated with tomato consumption. Thus, this study suggests a possible relationship between tomato microbial diversity and potential health benefits (lines 521-526).

Comment 13: L555: How the present study covers “detrimental effects” of “consuming spoiled tomatoes” on humans?

Response: We sincerely appreciate the reviewer's inquiry on the adverse effects of consuming spoiled tomatoes in humans. We believe that by incorporating a discussion on the negative effects of consuming spoiled tomatoes, we can give readers a more comprehensive understanding of the subject and its implications for human health. This is because the presence of pathogenic microorganisms was discovered in this study as tomato storage lengthened (lines 602-606). 

Comment 14: Conclusion section should contain the summary of the findings, not the speculation for the further research (L571-L584).

Response:  We greatly appreciate your valuable suggestion. While the conclusion of this study is adequately emphasized, we belief that providing additional recommendations would not only illuminate the study's limitations but also serve as a guiding beacon for researchers intending to extend the study's findings. This highlights the possibility for future study to build on the foundation we have established (lines 628-632).

Methods section comments

Comment 15: L118: was the surface of the fruits treated with detergent or another cleaning agent before the DNA extraction?

Response: The tomato fruit samples underwent no detergent or cleaning reagent treatment. They were aseptically transferred while wearing gloves into a zip lock bag and subsequently subjected to grinding using a stomacher apparatus. The observation has been included in the manuscript (Line 135-138).

Comment 16: L148: please specify the sequencing platform (MiSeq has different types) and read length.

Response:  Sequencing was conducted at MR DNA (www.mrdnalab.com, Shallowater, TX, USA) on an Illumina MiSeq platform (2x250bp), adhering strictly to the manufacturer's protocols. This observation can be found in the methodology section of the manuscript (Lines 166-168). Additionally, read lengths were 254 bp for bacteria and 240 bp for fungi (lines 218-219).

Comment 17: L151: Raw reads should be made public and mentioned in the manuscript.

Response: This is in process, when done demultiplexed raw sequence data will be deposited in the Sequence Read Archive (http://www.ncbi.nlm.nih.gov/sra)

Comment 18: L117: missing space.

Response: The observation has been amended in the manuscript

Reviewer 2 Report

Dear authors.

You have a great paper. However, before considering its publication I suggest some modifications:

1. Improve the introduction section showing information regarding previous studies.

2. Improve the quality and the letter size in the figures, in the actual MS it is not possible to review the information in the.

3. The use of a dendrogram could be useful to show the diversity of the microorganisms.

4. How the information shown in the paper is useful? could you choose some microorganisms to demonstrate their spoilage ability? 

5. Review the spaces near the references.

6. What can you explain about the relation between the different microbial groups?

7. Use italics in the names of the microorganisms.

The MS has moderate mistakes.

Author Response

Dear Editor,

Thank you for giving us the opportunity to submit a revised draft of our manuscript titled: Characterization of microbial diversity of two tomato cultivars through targeted next-generation sequencing 16S rRNA and ITS techniques to Microorganisms. We appreciate the time and effort that you and the reviewers have dedicated to providing your valuable feedback on our manuscript. We are grateful to the reviewers for their insightful comments on our paper. We have been able to incorporate changes to address most of the suggestions provided by the reviewers. We have highlighted the changes within the manuscript. We hope you find these revisions relevant, and we are ready to make further amendments if deemed necessary.

Thank you.

Here is a point-by-point response to the reviewers’ comments and concerns.

Reviewer #2

Comment 1: Improve the introduction section showing information regarding previous studies.

Response: Thank you for your insightful comment. We appreciate your feedback on the manuscript and have made the necessary changes accordingly. In the revised version of the text, we have incorporated recent data on the relationships between tomato plants and their microbiomes and further elaborated on prior studies on the microbial community associated with tomatoes (lines 74-105). These additions aim to provide a more comprehensive background information on this study and findings. Furthermore, the aim of the study is better aligned with the objectives of the research. We believe that these modifications will address the concerns you raised.

Comment 2: Improve the quality and the letter size in the figures, in the actual MS it is not possible to review the information in the.

Response: We sincerely apologize for any inconvenience caused by the presentation of our original Figures 3, 4, and 7. This was not our intention, and we deeply appreciate your attention to detail. We have taken your feedback into account and have made the necessary modifications to the figures to ensure clarity and proper visibility.

Comment 3: The use of a dendrogram could be useful to show the diversity of the microorganisms.

Response: We value your recommendation. Although a dendrogram could provide a valuable visual representation of microbial diversity, we chose bar charts in this instance to effectively illustrate the microbial diversity within tomato samples. Bar charts/graphs facilitate an easy comparison of relative abundances and are ideally suited for highlighting differences between samples. However, we recognise the potential advantages of incorporating dendrograms into future research in order to portray microbial relationships comprehensively.

Comment 4: How the information shown in the paper is useful? could you choose some microorganisms to demonstrate their spoilage ability?

Response: This observation has been elaborated in the discussion session of the manuscript (Lines 545-549, 571-568, 588-601).

Comment 5: Review the spaces near the references.

Response: This observation has been reviewed in the manuscript as suggested.

Comment 6: What can you explain about the relation between the different microbial groups?

Response: We agree that it would be useful to demonstrate the relationship between the different microbial groups; however, such an analysis is beyond the scope of our paper, which seeks to characterise the microbial communities of two tomato varieties: 'roma' and 'beefsteak' and how it changes during storage. Nonetheless, relationships such as Pathogenic Interactions, in which the spoilage capacity of Pantoa and Alternaria, to name a few, was demonstrated, and Microbial Succession, in which Proteobacteria and Ascomycota predominate, have been established.

Comment 7: Use italics in the names of the microorganisms.

Response: This observation has been reviewed in the manuscript.

Round 2

Reviewer 1 Report

I would to thank authors for the efforts to improve the manuscript, but it still requires multiple corrections. The quality of figures is still poor and hard to interpret. I recommend to attach the high resolution figures along with the manuscript.

The release of raw data is mandatory nowadays for microbial community studies, so I would insist on NCBI submission and extending the manuscript with SRA accession data.

Supplementary file contains unreadable Figure S2 and icons (!) not tables S1A...S5B. I think that preliminary submission check would get this issue. Please put tables in separate sheets or files.

The category names at Figures 3 & 4 are redundant and might be omitted. 

Table 1 was initially formatted well, but in revised version formatting was lost.

While the Discussion section was significantly extended, the Conclusion section remain the same. Future perspectives L608-L617 might be replaced to key findings interpretation.

Minor changes:

L568: "core fungal taxa of tomato": please compare with previous studies

L598: "it' to "It"

L585: ref. [80] doesn't cover any "allergic disorders"

Author Response

Dear Editor,

We thank you and the reviewer for the feedback. We have highlighted the changes within the manuscript as requested. Below is a point-by-point response to the reviewers’ comments.

Reviewer’s Comments

  1. The quality of figures is still poor and hard to interpret. I recommend to attach the high resolution figures along with the manuscript

Response: The quality of all the figures have been improved as suggested. We have also compiled the Figures in a separate PowerPoint slide for easy access and to allow for the Editorial Board to possibly adjust them as may be required for typesetting and editing. Many thanks.

  1. The release of raw data is mandatory nowadays for microbial community studies, so I would insist on NCBI submission and extending the manuscript with SRA accession data.

Response: The sequences have been successfully submitted, and the corresponding accession numbers have been provided in the manuscript (Lines 811-813).

  1. Supplementary file contains unreadable Figure S2 and icons (!) not tables S1A...S5B. I think that preliminary submission check would get this issue. Please put tables in separate sheets or files.

Response: This observation has been amended accordingly and can be found in the figure’s file. Tables are also separated in a different sheet.

  1. The category names at Figures 3 & 4 are redundant and might be omitted.

Response: This observation has been amended accordingly and can be found in both the manuscript and the figures file.

  1. Table 1 was initially formatted well, but in revised version formatting was lost.

Response:  Table 1 remains consistent with the original manuscript and the revised version Lines 224-225. However, this has also been included in the figures file for possible adjustment.

  1. While the Discussion section was significantly extended, the Conclusion section remain the same. Future perspectives L608-L617 might be replaced to key findings interpretation.

Response: This has been amended appropriately in the manuscript as advised. Many thanks.

  1. L568: "core fungal taxa of tomato": please compare with previous studies

Response: This observation was initially compared in the manuscript, they can be found in lines 744-750, 752-755 and 757-761.

  1. L598: "it' to "It"

Responses: This observation has been amended in the manuscript line 773.

  1. L585: ref. [80] doesn't cover any "allergic disorders"

Responses: This has been amended in the manuscript line 761 ref. [21].